# Intravenous treatment of choroidal neovascularization by photo-targeted nanoparticles

Yanfei Wang[1], Chi-Hsiu Liu[2], Tianjiao Ji[1], Manisha Mehta[1], Weiping Wang [1], Elizabeth Marino[1], Jing Chen[2] & Daniel S. Kohane[1]

Choroidal neovascularization (CNV) is the major cause of vision loss in wet age-related macular degeneration (AMD). Current therapies require repeated intravitreal injections, which are painful and can cause infection, bleeding, and retinal detachment. Here we develop nanoparticles (NP-[CPP]) that can be administered intravenously and allow local drug delivery to the diseased choroid via light-triggered targeting. NP-[CPP] is formed by PEG-PLA chains modified with a cell penetrating peptide (CPP). Attachment of a DEACM photo-cleavable group to the CPP inhibits cellular uptake of NP-[CPP]. Irradiation with blue light cleaves DEACM from the CPP, allowing the CPP to migrate from the NP core to the surface, rendering it active. In mice with laser-induced CNV, intravenous injection of NP-[CPP] coupled to irradiation of the eye allows NP accumulation in the neovascular lesions. When loaded with doxorubicin, irradiated NP-[CPP] significantly reduces neovascular lesion size. We propose a strategy for non-invasive treatment of CNV and enhanced drug accumulation specifically in diseased areas of the eye.

[1] Laboratory for Biomaterials and Drug Delivery, Department of Anesthesiology, Division of Critical Care Medicine, Boston Children's Hospital, Harvard Medical School, 300 Longwood Avenue, Boston, MA 02115, USA. [2] Department of Ophthalmology, Boston Children's Hospital, Harvard Medical School, Boston, MA 02115, USA. Correspondence and requests for materials should be addressed to D.S.K. (email: daniel.kohane@childrens.harvard.edu)

Retinopathy of prematurity, diabetic retinopathy, and vascular age-related macular degeneration (AMD) are the leading causes of blindness in infants, adults and the elderly in the US, respectively[1]. These diseases of varying etiology are characterized by the development of pathological neovascularization, which disrupts retinal structure and function, causing irreversible vision loss. Currently, the standard therapies for the treatment of neovascular eye diseases are laser photocoagulation and repeated intravitreal injections of antibodies against vascular endothelial growth factor[2,3]. They are effective in slowing or preventing neovascularization, but suffer from serious side effects: laser treatment inevitably destroys retinal tissue[4], and intraocular injections are unpleasant for the patients and bear risks of endophthalmitis and retinal detachment[5]. Less invasive means of administration of therapeutics, for example by intravenous injection, are therefore desirable. However, systemic administration of drugs often results in inadequate concentrations of drugs at the diseased site; this is particularly true of delivery to the back of the eye (retina and associated structures). Increasing drug levels at the target site by increasing the dose could lead to systemic toxicity. Recent advances in nanoparticle-based drug delivery systems (DDSs) provide opportunities to improve drugs' therapeutic effects[6]. DDSs that enable drug delivery to the back of the eye[7] are administered locally by intravitreal injection, or systemically. Systemic DDS can reach diseased sites due to the leaky vasculature in neovascular eye diseases[8,9], or by targeting

the ligand-modified DDS to specific antigens[10–13]. Such targeting is impeded by variability in the expression of ligand receptor at the diseased site and, and by the basal expression of certain target antigens (e.g., endoglin, integrin) in normal tissue[14].

Externally triggered targeting can enable drug delivery with high spatial and temporal resolution[15–19]. Light is especially attractive as the energy source for targeting the retina, since the eye is designed to admit light. We and others have demonstrated the possibility of using light to control targeting of nanoparticles to cells and tumors[20–23]. Here we design a system whereby nanoparticles (NPs) are administered intravenously, and are converted to a tissue-targeting state only upon irradiation in the eye (Fig. 1a). Our strategy would allow the targeted accumulation of drug to be triggered locally at the back of the eye, while minimizing drug deposition at off-target sites in healthy parts of the eye and in the rest of the body.

Photo-targeted nanoparticles are formed by self-assembly of a chemically modified poly(ethylene oxide)-poly(D,L-lactic acid) (PEG-PLA) block copolymer (Fig. 1b). The nanoparticles' surfaces are modified with Tat-C (48–57) cell penetrating peptide (CPP) as the targeting moiety due to its high cellular uptake[24]. The biological activity of the peptide is reversibly inactivated by covalent binding to a photocleavable caging group, 7-(diethylamino) coumarin-4-yl]methyl carboxyl (DEACM), which is selected for its high photocleavage efficiency and relatively long (400 nm) absorption wavelength (low phototoxicity)[25]. Upon

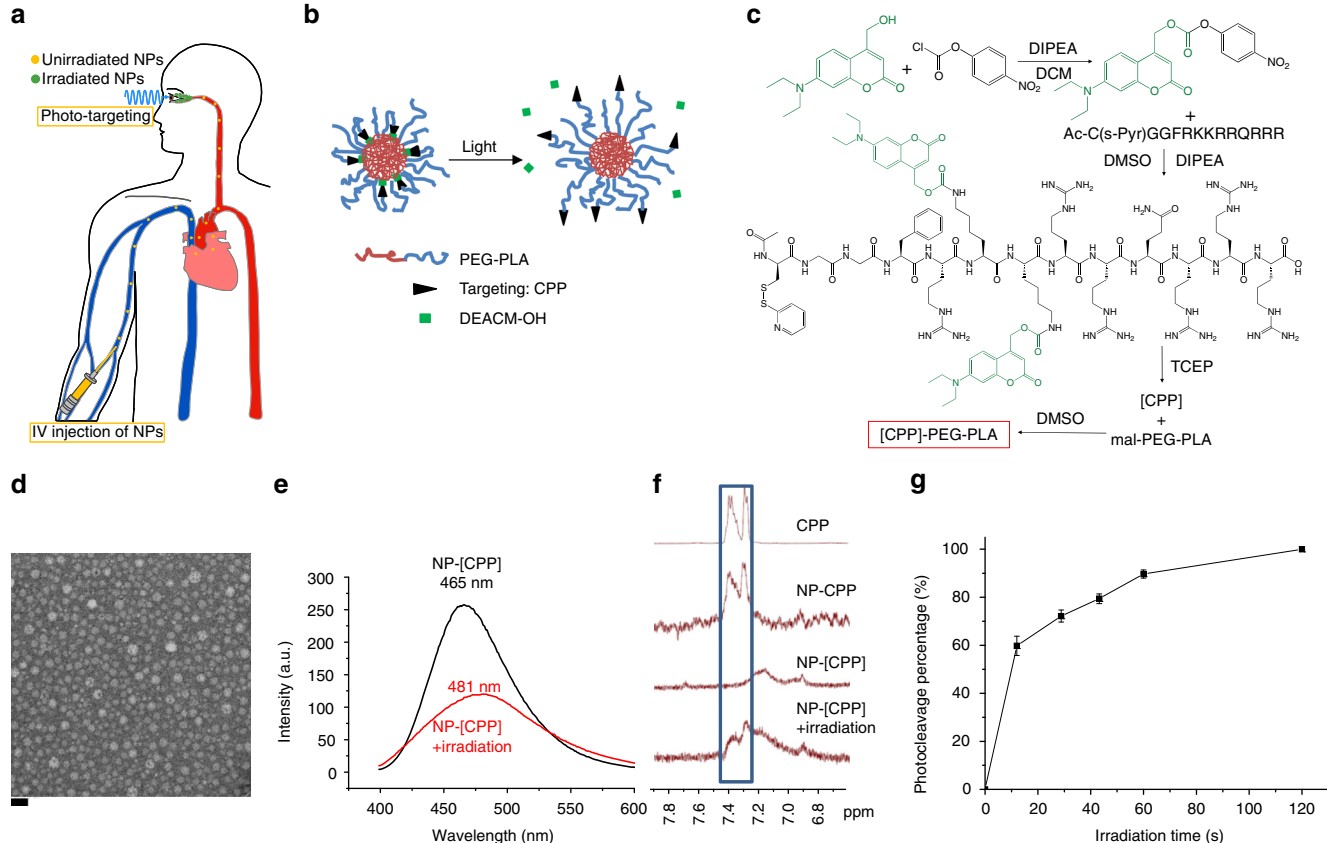

**Fig. 1** Preparation and characterization of phototargeted nanoparticles. **a** Phototargeting intravenously administered nanoparticles to choroidal neovascularization. **b** Schematic of light-triggered activation of the nanoparticle. **c** Synthesis of the polymer chain functionalized with caged CPP ([CPP]). **d** Transmission electron microscopy (TEM) image of NP-[CPP]. The scale bar is 50 nm. **e** Fluorescence emission spectra of NP-[CPP] and NP-[CPP] irradiated for 1 min (50 mW cm$^{-2}$, 400 nm) in PBS, the emission maxima are labelled. **f** $^1$H NMR spectra of free CPP and different nanoparticles in D$_2$O, with the signature phenylalanine proton peaks highlighted in the blue rectangle. NP-CPP is the nanoparticle formed from CPP-PEG-PLA and mPEG-PLA (1:4 weight ratio). Irradiation was with a 400 nm LED for 1 min at 50 mW cm$^{-2}$. **g** Photocleavage of NP-[CPP] in PBS (0.5 mg mL$^{-1}$), as determined by HPLC (detected at 390 nm absorbance), after continuous irradiation (50 mW cm$^{-2}$, 400 nm) (data are means ± SD; $n = 4$ independent experiments)

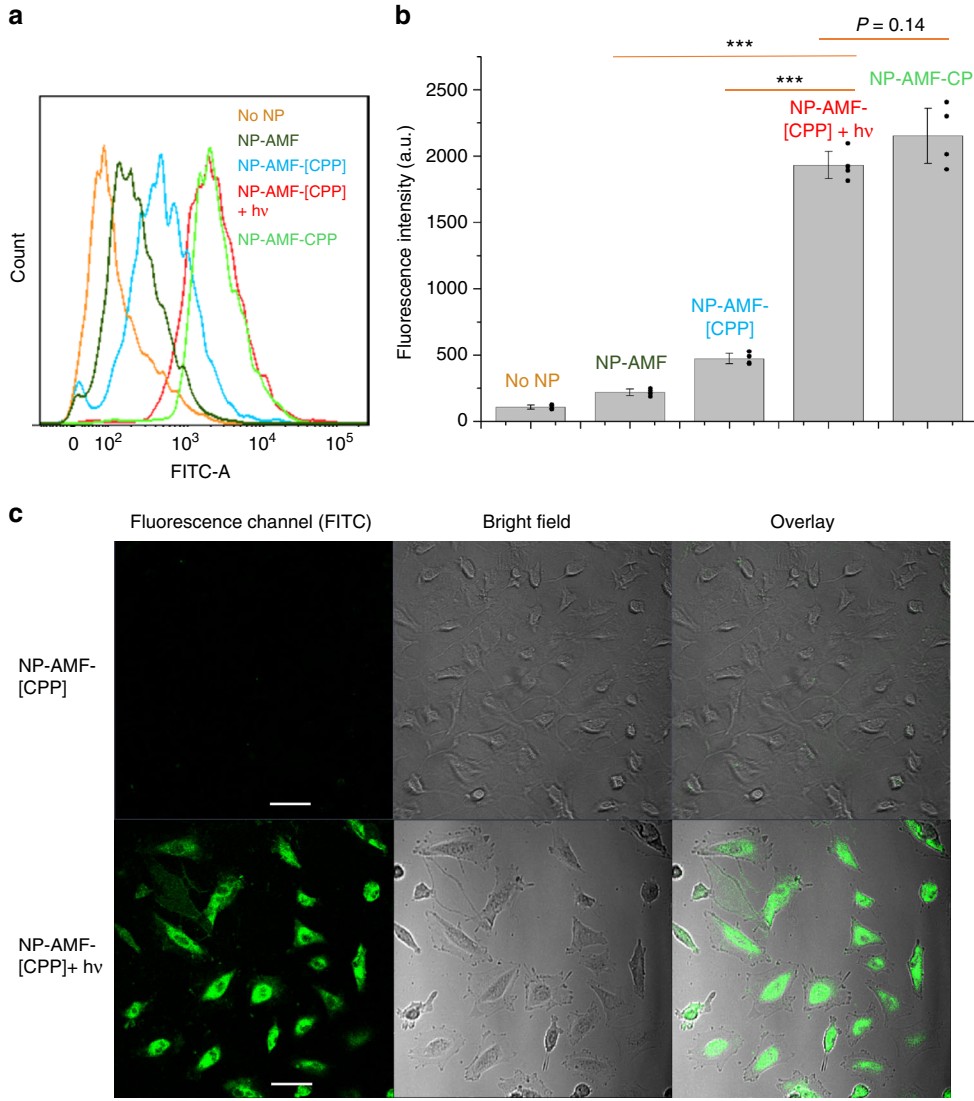

**Fig. 2** Light-triggered cell uptake of nanoparticles. **a** Representative flow cytometry of FITC fluorescence within HUVEC cells treated with different nanoparticles or without nanoparticle. **b** Quantitation (mean of four median values of fluorescence intensity) of flow cytometric analyses (such as the one in panel A) of HUVEC uptake of nanoparticles. Data are means ± SD ($n = 4$ independent experiments). ***$P < 0.001$ (unpaired $t$-test). **c** Representative confocal microscopic images of HUVEC uptake of nanoparticles. The scale bar is 20 μm

irradiation, the caging group would be removed by bond cleavage so that the peptide could readily bind to nearby cells. The DEACM-CPP functionalized nanoparticles could then enhance the accumulation of drugs at the diseased site and minimize off-target drug delivery. Importantly, this approach would obviate the need for intraocular injections with their attendant risks, and improve patient compliance.

## Results

**Synthesis and characterization of photo-targeted nanoparticles.** Figure 1c shows the synthesis of caged CPP ([CPP])-functionalized polymer chain, the details of which are provided in Supporting Information. In brief, [CPP] was synthesized by linking DEACM to the amine group on peptide side chains via nucleophilic substitution (Supplementary Fig. 1). The [CPP] with cysteine at the N-terminus (amino acid sequence acetyl-CGGFRKKRRQRRR) was then conjugated to the PEG end of maleimide-modified PEG-PLA via maleimide-thiol coupling. Photo-targeted nanoparticles were made by the thin-film

hydration method from [CPP]-PEG-PLA and methoxy PEG-PLA (mPEG-PLA) (1:4 weight ratio). The resulting micelles, referred to as photo-targeted nanoparticles and abbreviated as NP-[CPP] (Fig. 1b), had a hydrodynamic diameter of 19.0 ± 2 nm (means ± SD; $n = 4$, Fig. 1d and Supplementary Fig. 2).

We hypothesized that the hydrophobic DEACM groups would localize in the PLA core of the nanoparticle, and the photocleavage reaction would release the more hydrophilic DEACM-OH into the aqueous environment. This hypothesis was supported by the fact that the fluorescence spectrum of NP-[CPP] solution showed a red shift and decrease in the emission intensity upon irradiation with 400 nm LED light (Fig. 1e). The red shift (from 465 to 481 nm in maximum emission wavelength) was attributable to the increased polarity of DEACM's environment and the decrease in intensity to the quenching of fluorescence by water[26]. Further evidence indicating the presence of the CPP in the hydrophobic core was obtained by proton nuclear magnetic resonance (1H NMR) spectroscopy. The 1H NMR spectrum of NP-[CPP] did not show peaks of the phenyl protons from phenylalanine in the range of 7.25–7.45 ppm,

because of the restricted mobility of the phenyl protons within the PLA cores of the nanoparticles[27]. Irradiation resulted in the appearance of those peaks, confirming that the phenylalanine in NP-[CPP] was located in the PLA core and photocleavage led to its translocation to the surface (Fig. 1f).

To measure the rate of phototriggered release of DEACM-OH from NP-[CPP], a quartz cuvette containing 1 mL of 0.5 mg mL$^{-1}$ NP-[CPP] solution was continuously irradiated with 400 nm LED light at 50 mW cm$^{-2}$. At predetermined time points, DEACM was separated from the nanoparticle solution by centrifugation and DEACM-OH content was determined (Fig. 1g) by high-performance liquid chromatography (HPLC) ($\lambda = 390$ nm). $89.7 \pm 1.7\%$ of DEACM−OH was released after 1 min.

**Nanoparticle uptake by cells.** Cellular uptake of nanoparticles by human umbilical vein endothelial cells (HUVECs) was studied by flow cytometry and confocal microscopy. Nanoparticles were labeled by addition of PEG-PLA copolymer to which the hydrophilic dye 4'-(aminomethyl) fluorescein (AMF; excitation 491 nm; emission 524 nm) was covalently bound. The weight percentage of AMF-PEG-PLA in the nanoparticles was 10%.

To determine the proportion of peptide-polymer conjugate to use in subsequent experiments, HUVECs were incubated with nanoparticles containing varying proportions of CPP-PEG-PLA or [CPP]-PEG-PLA. HUVEC uptake of nanoparticles increased with the proportion of CPP-PEG-PLA to a maximum at 40% w/w, then decreased at higher proportions (Supplementary Fig. 3a). We then studied the effect of caging on cell uptake at the two loadings with the greatest uptake when uncaged. Nanoparticles with 40% w/w [CPP]-PEG-PLA were taken up by cells to a greater extent than were nanoparticles with 20% w/w [CPP]-PEG-PLA (Supplementary Fig. 3b). From the flow cytometric analyses (Supplementary Fig. 3), the ratio of cell-associated NP fluorescence in cells exposed to NPs with 20% CPP to NPs with 20% [CPP] was 4.7; that of NPs with 40% CPP to those with 40% [CPP] was 3.3. Given that off-target delivery remains a dominant problem even with targeted systems[22] (also see biodistribution data below), we therefore used nanoparticles with 20% w/w [CPP]-PEG-PLA in subsequent experiments.

HUVECs were incubated for 30 min with the following nanoparticles containing AMF-PEG-PLA: unmodified mPEG −PLA nanoparticles (NP-AMF), nanoparticles modified with CPPs (NP-AMF-CPP), NP-AMF-[CPP] without irradiation, and NP-AMF-[CPP] irradiated with a 400 nm LED (50 mW cm$^{-2}$, 1 min). Cell-associated AMF fluorescence was measured by flow cytometry. HUVECs incubated with NP-AMF-CPPs exhibited 9.9-fold greater fluorescence than those exposed to peptide-free nanoparticles (Fig. 2a and b), which demonstrated the ability of CPP to bind nanoparticles to cells. NP-AMF-[CPP] exhibited little cell-associated fluorescence, suggesting that the caging strategy prevented ligand-mediated NP-cell interaction. Irradiation with a 400 nm LED (50 mW cm$^{-2}$, 1 min) increased cellular uptake to levels comparable to those with NP-AMF-CPP. These results confirmed that the DEACM caging group could be cleaved from NP-AMF-[CPP] by irradiation, which revealed CPP on the nanoparticle surface and enabled cellular uptake. The size and granularity (internal complexity) of cells remained the same for all groups (Supplementary Fig. 17).

Light-controlled micelle uptake was further confirmed by confocal laser scanning microscopy. Cells incubated with NP-AMF-CPP showed strong fluorescence, and those incubated with NP-AMF showed negligible fluorescence. (Supplementary Fig. 19) Irradiation with a 400 nm LED (50 mW cm$^{-2}$, 1 min) induced cell uptake of NP-AMF-[CPP] by HUVECs, whereas the uptake of non-irradiated NP-AMF-[CPP] was negligible (Fig. 2c).

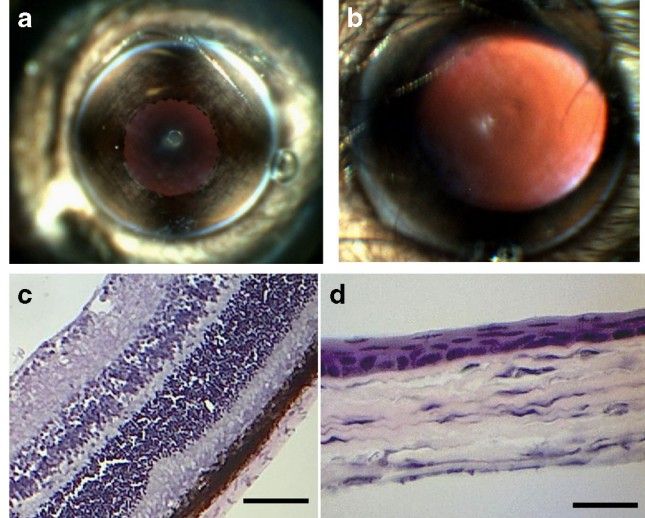

**Fig. 3** Murine ocular tissue reaction to 400 nm LED irradiation at 50 mW cm$^{-2}$ for 5 min. **a** Photograph showing clear cornea with unobstructed view of underlying iris. **b** Photograph showing clear cornea and lens with unobstructed view of retina through dilated pupil. **c** Photomicrograph of hematoxylin-eopsin stained section of retina, showing no detectable injury. The scale bar is 50 µm. **d** Photomicrograph of hematoxylin-eopsin stained section of cornea, showing no detectable injury. The scale bar is 25 µm

**Cytotoxicity and biocompatibility of treatments.** The cellular target of treatment in neovascularization is the endothelium lining the neovessels; we therefore tested the cytotoxicity of nanoparticles and/or irradiation in HUVECs by MTS assay. HUVECs were exposed to irradiation (400 nm LED for 1 min at 50 mW cm$^{-2}$), or incubated with 0.5 mg mL$^{-1}$ NP-[CPP] overnight with irradiation (1 min, at the beginning of incubation) or without. NPs were removed before the MTS assay. Non-treated HUVECs were used as controls. All three groups showed high cell viability (Supplementary Fig. 4).

The safety of the irradiation conditions was further studied in vivo. Under isoflurane anesthesia, the eyes of C57BL/6 mice were irradiated with 400 nm LED for 5 min at 50 mW cm$^{-2}$. No abnormalities such as corneal clouding or signs of cataracts were observed with a fundus camera (see Methods) within 48 h after irradiation (Fig. 3). Tissue sections (2 weeks after irradiation) stained with hematoxylin and eosin (H&E) revealed normal histology in irradiated cornea and retina, and no detectable difference between irradiated and non-irradiated eyes (comparing Fig. 3c, d and Supplementary Fig. 5).

**Light-triggered targeting in vivo in mouse CNV model.** The laser-induced mouse model of CNV[28] was used to investigate nanoparticle phototargeting in vivo. In brief, CNV was induced by laser photocoagulation-induced (532 nm, 0.24 W, 0.07 s) rupture of Bruch's membrane of C57BL/6 mice. Four laser burns indicated by the development of vapor bubbles in Bruch's membrane were induced per eye around the optic disc (approximately 0.5–1 mm from the optic nerve). Fundus fluorescein angiography (FA) was used to monitor the development of the vascularity associated with CNV (Supplementary Fig. 6).

One week after laser treatment, groups of mice with induced CNV were injected intravenously (IV) with 200 µL (5 mg mL$^{-1}$) of AMF-PEG-PLA-labeled nanoparticles in four groups: NP-AMF, NP-AMF-CPP, NP-AMF-[CPP], and NP-AMF-[CPP] with irradiation (NP-AMF-[CPP]+hv) immediately (30 s) after IV injection. Thirty seconds after IV injection, the fluorescent nanoparticles were observed in the mouse fundus by FA and

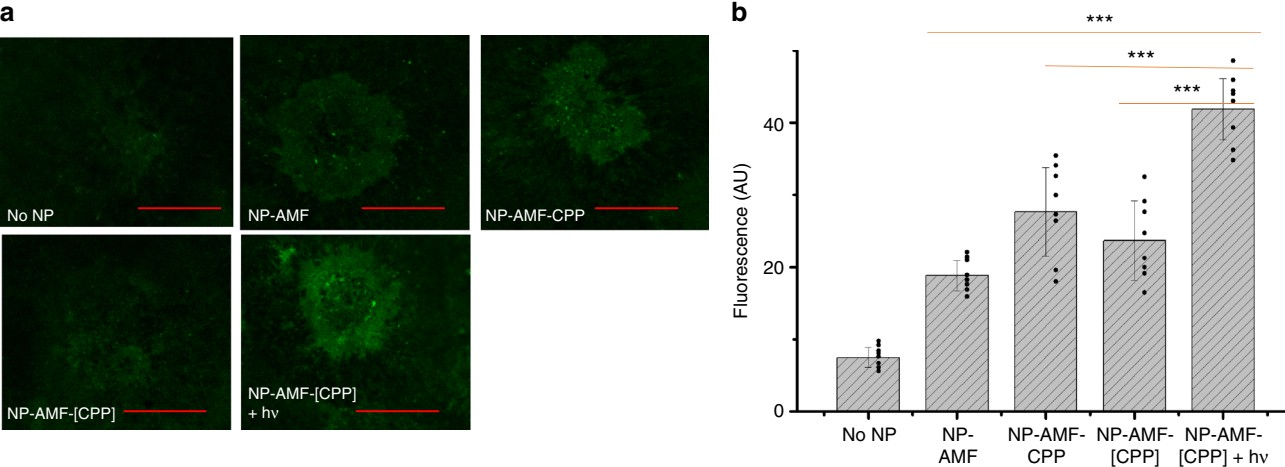

**Fig. 4** Light-triggered targeting of CNV in vivo. **a** Representative (of 8) fluorescent images of flat-mounted choroid 24 h after injection with NPs. The scale bar is 100 μm. **b** Quantification of the intensity of fluorescent neovessels from images in **a**, normalized by the lesion size. Data are means ± SD ($n = 8$ lesions) ***$P < 0.001$ (unpaired $t$-test)

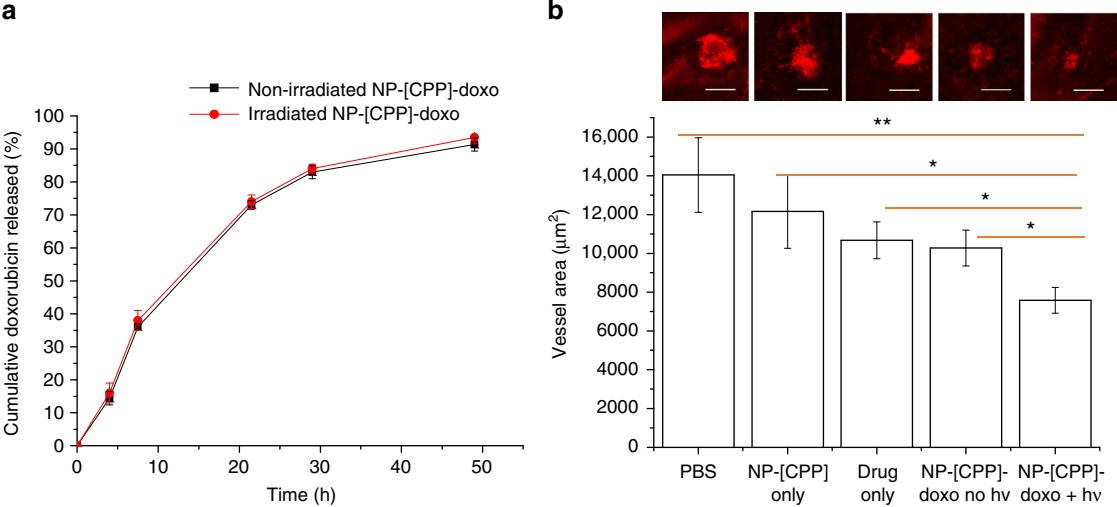

**Fig. 5** Treatment with NP-[CPP]-doxo in mouse CNV model. **a** Cumulative doxorubicin release (as % of total amount loaded) from NP-[CPP]-doxo at 37 °C in vitro by dialysis, with or without irradiation (400 nm LED for 1 min at 50 mW cm$^{-2}$) at $t = 0$. Data are means ± SD ($n = 4$ independent experiments). **b** Representative isolectin GS IB$_4$-stained CNV images (the scale bar is 100 μm) and mean CNV lesion area from laser-induced CNV mice treated with PBS, NP-[CPP], doxo, NP-[CPP]-doxo, and NP-[CPP]-doxo with irradiation. Error bars show standard error of the mean. $n = 28$ lesions. *$P < 0.05$, **$P < 0.005$ (unpaired $t$-test). (See Supplementary Fig. 14 for graph of individual data points and Supplementary Table 1 for the standard deviations.)

the intense fluorescence persisted for 5 min (Supplementary Video 1, 2). Fluorescence was brightest during the first 4 min after IV injection of nanoparticles, in both retinal blood vessels and laser-induced lesions (Supplementary Fig. 7). Fluorescence in the retinal blood vessels was still visible in vivo 8 h after nanoparticle injection, but not 24 h after injection (Supplementary Fig. 18). Microscopy of the flat-mounted retina 24 h after IV injection (Supplementary Fig. 8) revealed no fluorescence in the retinal vessels.

To evaluate the accumulation of nanoparticles in the neoformed CNV lesions, mice were euthanized 24 h after nanoparticle injection, and their eyes were enucleated for laser scanning confocal microscopy of flat-mounted choroids (Fig. 4). Higher fluorescence intensity was observed in the eyes of mice in the NP-[CPP]+hν group than in the others (NP, NP-CPP and NP-[CPP] without irradiation), confirming that photo-targeting enhanced nanoparticle accumulation in the CNV areas. Minimal

fluorescence was observed in choroidal flat-mounts of healthy eyes of mice injected with nanoparticles (Supplementary Fig. 9), suggesting that the leaky vasculature in CNV plays an important role in nanoparticle accumulation. This observation was consistent with the previously reported enhanced permeability and retention (EPR)-like effect in CNV[8].

Biodistribution of nanoparticles was determined by measuring the fluorescence intensity in organs collected 24 h after nanoparticle injection. When AMF was extracted from the same organs and the fluorescent content was measured (Supplementary Fig. 10a), it was found in all organs, with the greatest concentration in liver. Fluorescence was undetectable by this approach in all choroids, even though fluorescence was seen on microscopy, possibly due to the small amounts of material involved (the entire choroid weighed ~1 mg). Therefore, the mean fluorescence intensity in the choroids (from microscopy, Fig. 4b) was used as the metric for AMF accumulation in the eye. We

found that the ratio of AMF accumulation in the eye to that in other organs in the NP-AMF-[CPP]+hv group was roughly twice (1.7–2.4-fold, depending on the organ) that in the NP-AMF-[CPP] group (Supplementary Fig. 10b).

**Therapeutic effects of phototriggered activation of nanoparticles in CNV.** The mouse model of laser-induced choroidal neovascularization mimics the neovascular aspect of wet AMD, and has been used extensively in studies of that disease[29,30]. However, unlike the chronic development of wet AMD in humans, the laser-induced mouse CNV model is acute–the optimum time point to assess neovessels in this model is between day 7 and day 14; after 14–21 days, spontaneous regression begins and lesion size decreases[29]. Our dosing regimen was determined accordingly: treatment was initiated on day 7 when the lesions had formed, and its effects were assessed before spontaneous regression would begin on day 14.

The drug doxorubicin (doxo) which inhibits CNV when injected intraocularly[31] was converted to the free-base form and then encapsulated in the nanoparticles by a one-step process where the drug was co-dissolved with polymers in an organic solvent and under reduced pressure formed a thin film, which was then hydrated to form nanoparticles. The drug loading was $0.2 \pm 0.03$ mg of doxorubicin per 10 mg of nanoparticles; the loading efficiency was $39.0 \pm 4.1\%$. In vitro release of doxorubicin was assessed by dialyzing 500 µL of NP-[CPP] encapsulating doxorubicin (NP-[CPP]-doxo) against 14 mL of phosphate-buffered saline (PBS). $91.3 \pm 2.0\%$ of doxorubicin was released in the first 48 h, and release was complete by 7 days (Fig. 5a).

In the in vivo therapeutic study, 200 µL of PBS containing 1 mg of NP-[CPP]-doxo encapsulating 0.02 mg doxorubicin were injected via tail vein 1 week after photocoagulation (day 1 of treatment), and additional injections were given on day 3 and day 5. (Given the release kinetics of the nanoparticles, this would constitute ~1 week of treatment[29].) Thirty seconds after nanoparticle injection, the mouse eyes were irradiated with a 400 nm LED for 3 min at 50 mW cm$^{-2}$ (NP-[CPP]-doxo+hv). Analogous experiments were done with PBS, free drug, NP-doxo, and non-irradiated NP-[CPP]-doxo. (The three drug-loaded groups contained equal doses of doxorubicin.) On day 7, mice were euthanized, the size of CNV lesions were assessed by fluorescent imaging of choroidal flat-mounts stained with isolectin GS-IB$_4$ (which labels endothelial cells). Mice treated with NP-[CPP]-doxo+hv showed a 46.1% reduction in neovessel area compared to the group treated with PBS. The free doxorubicin group and that with NP-[CPP]-doxo without irradiation showed 24.0% and 26.8% reductions in neovessel area, respectively, approximately half of the inhibitory effect of NP-[CPP]-doxo with irradiation (Fig. 5).

**Tissue reaction.** Histological study of organs including the heart, kidney and spleen revealed normal histology and did not show any detectable differences between the treated groups and saline-injected animals (Supplementary Fig. 11). In all groups including saline-injected controls, there were foci of mild acute interstitial inflammation in predominantly normal lungs and of chronic inflammation in the liver (Supplementary Fig. 12). These changes are attributed to infection/inflammation related to housed animals[32] because they were seen across all groups including the untreated controls.

## Discussion

We have developed an intravenously injected, photo-targeted treatment regimen for CNV and demonstrated its efficacy in the standard mouse CNV model. We showed that phototargeting can enhance nanoparticle accumulation in the abnormal vessels in CNV, and can significantly enhance treatment. Notably, the inhibitory effect of NP-[CPP]-doxo plus irradiation on CNV growth was comparable to that of locally administered drug formulations reported previously[31,33,34]. The intensity and duration of irradiation used in our study did not cause damage to the mouse eyes, and NP-[CPP]-doxo did not cause tissue toxicity.

Further developments of this proof-of-principle system could be designed to provide better therapeutic efficacy by changing the drug used, and/or drug dosage and irradiation parameters. The dosing regimen we used was dictated in part by the characteristics of the mouse model of laser-induced CNV, wherein lesions develop over 7 days, and regress spontaneously from 14 days on[29]. However, the disease that this model mimics – the exudative form of AMD – is chronic. Consequently, the best therapeutic regimen in humans might be more extensive (for example, repeated weekly IV injections), perhaps with even more pronounced benefits. Moreover, the formulation used here could be further improved with respect to drug loading, nanoparticle concentration, duration of infusion, and perhaps other nanoparticle characteristics (size, ligand type, etc.). The irradiation itself could perhaps be modified in intensity and duration to enhance nanoparticle uptake in the eye as well as patient tolerance. The specific payload of the NP-[CPP] could also be modified, since they could encapsulate a variety of drugs, singly or in combination. The retention and clearance of doxorubicin in mouse organs could be monitored by fluorimetry (of doxorubicin) of organ extracts at different time points following intravenous injection of doxorubicin-containing NPs. (This might also be possible in CNV lesions at higher doxorubicin loadings.) Such information may be helpful in optimizing the dosing regimen for treating CNV by the intravenous route, especially if further modification of the phototargeted NPs provided more extended release of the encapsulated therapeutics which would allow for less frequent dosing. The enhanced effect in the phototriggered group in our study is attributable to the enhanced accumulation of drug, as evidenced indirectly by enhanced nanoparticle accumulation in the CNV lesion. We cannot exclude a possible therapeutic contribution from the photo-triggered accumulation of NP-CPP.

This work contributes to the small but growing body of evidence that photo-targeted (and other targeting) strategies can increase local drug deposition with resulting therapeutic benefits, and extends it to non-cancer conditions[16,22]. (It is not clear that such benefits accrue in conditions where some process such as EPR does not facilitate NP uptake into the diseased tissue). As has been correctly pointed out[35], the amounts of drug involved are quite small as a percentage of the total dose delivered, as further evidenced by the fact that our biodistribution studies could not detect any extractable fluorescent dye in eye tissue. Nonetheless, those small amounts are clearly effective, as we have demonstrated here and in a tumor model[22]. We have argued that what matters is perhaps not the absolute amount delivered, but whether the amount at the target site can be increased and in particular whether it can be increased in proportion to the amounts delivered to off-target sites (i.e. whether the therapeutic index can be improved)[16].

Currently, the primary treatment of CNV is monthly intravitreal injections of vascular endothelial growth factor (VEGF) inhibitors[36]. Frequent intravitreal injections pose risks such as bleeding, infection, retinal detachment and degeneration[37], and patients find them unpleasant, which may be a major cause of non-compliance[38]. The nanoparticulate therapy described here can be triggered by safe, inexpensive and portable LEDs, potentially enabling broad clinical use. This system could mitigate the need for invasive intraocular injections in treating ocular diseases, specifically other neovascular retinal diseases.

## Methods

**Reagents**. Chemicals were purchased from Sigma-Aldrich (Missouri, USA) and used without further purification unless otherwise stated. Poly(D,L-lactic acid) (2000)-poly(ethylene oxide)(3000)-N- hydroxysuccinimide (PLA-PEG-NHS) and PLA(2000)-methoxy PEG (mPEG, 2000) (PLA- mPEG) were ordered from Advanced Polymer Materials (Montreal, Canada). 7-diethylamino-4- hydroxymethylcoumarin was bought from INDOFINE Chemical Company (New Jersey, USA). The CPP (amino acid sequence Ac-{Cys(2-pyridinesulfenyl)} GGFRKKRRQRRR) was purchased from GL Biochem Shanghai LTD (Shanghai, China). Human umbilical vein endothelial cells (HUVECs) and endothelial cell growth media kits (EGMTM-2 BulletKit, Catalog # CC-3162) were purchased from Lonza (New Jersey, USA). The CellTiter® 96 AQueous One Solution Cell Proliferation Assay solution was purchased from Promega (Wisconsin, USA). Other cell culture agents were purchased from Life Technologies (Now York, USA).

**Synthesis of caged peptide ([CPP])**. (7-(diethylamino)-2-oxo-2H-chromen-4-yl) methyl (4- nitrophenyl) carbonate (DEACM-carbonate) was synthesized by reacting 7-diethylamino-4-hydroxymethylcoumarin (DEACM-OH, 0.6 mmol) with 4-nitrophenyl chloroformate (2.4 mmol) in the presence of N,N-Diisopropylethylamine (DIPEA, 2.4 mmol) in dichloromethane (DCM)[21]. DEACM-carbonate was then reacted with the CPP as follows: 56.7 mg of DEACM-carbonate (0.137 mmol) was dissolved in 600 μL of DCM. N,N-Diisopropylethylamine (DIPEA, 23.94 μL, 0.137 mmol) and 60 mg of CPP (-SH protected by 2-pyridinesulfenyl) (0.034 mmol) dissolved in 400 μL of DMF. DEACM-carbonate solution was then added dropwise to the CPP solution. The mixture underwent mechanical agitation for 24 h, then was separated by HPLC (C4 column, H8 214TP52, 2.1 × 150 mm, from Vydac, Hesperia, CA). The mobile phase consisted of solvent A (0.05% TFA in H$_2$O) and B (0.043% TFA, 80% ACN in H$_2$O). Ten percent of solvent B was used as the mobile phase from 0 to 10 min, then the percentage of B in the mobile phase was linearly increased to 100% from 10 to 55 min. The flow rate was 0.3 mL min$^{-1}$.

Purified 2-pyridinesulfenyl protected caged CPP was then mixed with TECP (Thermo Scientfic, Pierce™ Immobilized TCEP Disulfide Reducing Gel) to remove the protection group and purified with HPLC (same method as above) to yield [CPP]. The mass spectrum of [CPP] shows only peaks corresponding to CPP modified with two photo-caging groups (Supplementary Fig. 1).

**Synthesis of polymer-peptide conjugates**. To synthesize [CPP]-PEG-PLA, 20 mg of maleimide-PEG-PLA and 10 mg of [CPP] were dissolved in 500 μL DMSO-d$_6$. The mixture was shaken at room temperature and $^1$H NMR was used to monitor the reaction until the methine proton peak from maleimide disappeared (Supplementary Fig. 13). The reaction mixture was diluted with H$_2$O, placed in a Spectra/Por® 6 dialysis membrane (molecular weight cutoff, MWCO: 3500 Da) and dialyzed against 4 changes of 2 L of distilled water at 4 °C. After 2 days of dialysis, the dialyzed solution was lyophilized.

To synthesize AMF-PEG-PLA, 20 mg of NHS-PEG-PLA and 4 mg of 4'-(aminomethyl)fluorescein (Thermo Fisher Scientific) were dissolved in 300 μL DMSO. A total of 0.6 μL of DIPEA was added, and the mixture was shaken for 5 h at room temperature. The mixture was diluted with H$_2$O, then dialyzed (as above) and lyophilized.

**Preparation of polymeric nanoparticles**. To prepare NP-[CPP], [CPP]-PEG-PLA (2.0 mg) and PLA-mPEG (8.0 mg) were co-dissolved in 5 mL of chloroform. Rotary evaporation was used to slowly evaporate the solvent at 45 °C. The dried polymer film was hydrated with 2 mL of PBS at 60 °C. For other nanoparticles, the same procedure was used except that different compounds were added for each type of nanoparticle. For NP: mPEG (10.0 mg); NP-CPP: CPP-PEG-PLA (2.0 mg) and mPEG-PLA (8.0 mg). For NP-AMF: AMF-PEG-PLA (1.0 mg) and mPEG-PLA (9.0 mg). For NP-AMF-[CPP]: [CPP]-PEG-PLA (2.0 mg), AMF-PEG-PLA (1.0 mg) and mPEG-PLA (7.0 mg). For NP-AMF-CPP: CPP-PEG-PLA (2.0 mg), AMF-PEG-PLA (1.0 mg) and mPEG-PLA (7.0 mg).

Characterization of micelles by transmission electron microscopy (Supplementary Fig. 16): a 10 μL aliquot of the nanoparticle solution was deposited on a copper grid coated by a carbon film. After 2 min, excess solution was blotted by a filter paper. The sample was dried at room temperature and then imaged on a Tecnai G$^2$ Spirit BioTWIN transmission electron microscope (FEI company, OR, USA) operating at 80 kV.

Characterization of nanoparticles by dynamic light scattering: the size of nanoparticles was measured with a Delsa Nano C particle analyzer (Beckman Coulter, CA, USA) (Supplementary Table 2). Nanoparticle solution (100 μL) was put into a disposable cuvette (Eppendorf UVette) and tested at 25 °C with the accumulation times of 70. Each sample was tested at least three times. The hydrodynamic diameter was calculated by averaging the repeated measurements of diameters.

The zeta potential of nanoparticles (Supplementary Table 2) was measured by a PALS Zeta Potential Analyzer (Brookhaven Instruments Corp.). A total of 0.1 mg mL$^{-1}$ NPs (800 μL) in 5 mM phosphate buffer was tested at 25 °C for at least four times. When incubated with 10% FBS solution, the NPs did not form aggregates over 48 h.

**Photocleavage of DEACM-OH from NP-[CPP]**. The mechanism of the photocleavage reaction is shown in Supplementary Fig. 20. To measure the rate of phototriggered release of DEACM from NP-[CPP], a quartz cuvette containing 1 mL of NP-[CPP] solution (0.5 mg mL$^{-1}$) was irradiated under an 11-mm LED light (400 nm) collimator with a multi-channel Universal LED controller (Mightex Systems, CA, USA). The temperature of the solution was controlled at 37 °C in a t50/Eclipse cuvette holder with a TC 125 temperature controller (Quantum Northwest, WA, USA). The LED irradiance was measured with a PM100USB Power and Energy Meter (ThorLabs, NJ, USA). At each irradiation time point, the solution was put in an Amicon® Ultra centrifugal filter (MWCO: 50000 Da) and centrifuged at 3082 × g for 20 min. The filtrate was analyzed by RP-HPLC ($\lambda = 390$ nm) with a Poroshell 120 EC-C18 column. At time 0, there was no detectable DEACM-OH in the NPs by HPLC. After irradiation, NPs were separated from the solution by centrifugal filtration and then dissolved in acetonitrile and analyzed by HPLC. There was no detectable DEACM in the NPs by HPLC analysis.

**Flow cytometry**. Human umbilical vein endothelial cells (HUVECs) were purchased from Lonza (Catalog #: CC-2519, New Jersey, USA). Cells were cultured in cell growth media in a humidified atmosphere with 5% CO$_2$ at 37 °C in a 48-well plate at a density of 40,000 cells per well. After overnight incubation, the growth media were replaced with fresh media containing different nanoparticles at a concentration of 0.4 mg mL$^{-1}$, in the following groups: NP-AMF, NP-AMF-CPP, NP-AMF-[CPP], and NP-AMF-[CPP] with irradiation (400 nm, 50 mW cm$^{-2}$, 1 min). After 30 min of incubation at 37 °C, the cells were washed with PBS twice and detached with 150 μL of 0.25% Trypsin-EDTA solution. The cells were suspended with 350 μL of trypsin neutralizing solution (TNS) and transferred into BD Falcon round-bottom tube (BD Bioscience, NJ, USA). The flow cytometry was run on BD LSR Fortessa cell analyzer (BD Bioscience, NJ, USA).

**Confocal laser scanning microscopy**. Cells were seeded on a 35-mm glass bottom dish with collagen coating (MatTek Corporation, MA, USA) at a density of 250,000 cells per well. After overnight incubation, the growth media were replaced with the fresh media containing different nanoparticles at a concentration of 0.4 mg mL$^{-1}$, in the following groups: NP-AMF, NP-AMF-CPP, NP-AMF-[CPP], and NP-AMF-[CPP] with irradiation (400 nm, 50 mW cm$^{-2}$, 1 min). After 30 min of incubation at 37 °C, the cells were washed with PBS twice and imaged by confocal microscopy (488 nm, Zeiss LSM 710).

**Cytotoxicity analysis**. Cell viabilities were evaluated with an assay of mitochondrial metabolic activity (the MTS assay), CellTiter 96 AQueous One Solution Cell Proliferation Assay (Promega Corp.), that uses a tetrazolium compound [3-(4,5-dimethyl- 2-yl)-5-(3-carboxymethoxyphenyl)-2-(4-sulfophenyl)-2H-tetrazolium, inner salt (MTS)] and an electron coupling reagent (phenazine ethosulfate). HUVEC were incubated with CellTiter 96 AQueous One Solution for 120 min at 37 °C. The absorbance of the culture medium at 490 nm was immediately recorded with a 96-well plate reader. The quantity of formazan product (converted from tetrazole) as measured by the absorbance at 490 nm is directly proportional to cell metabolic activity in culture.

**Production of doxorubicin free-base**. Ten milligrams of doxorubicin hydrochloride salt was dissolved in H$_2$O. Ten microliters of trimethylamine was added to the solution under mechanical agitation. The mixture was centrifuged at 3901 × g for 15 min, and the precipitate was washed with water three times, then dried under vacuum.

**Loading efficiency of doxorubicin in NP-[CPP]**. To prepare NP-[CPP]-doxo, [CPP]-PEG-PLA (2.0 mg), mPEG-PLA (8.0 mg) and doxorubicin (0.5 mg) were co-dissolved in 5 mL of chloroform. Rotary evaporation at 45 °C was used to slowly remove the solvent. The dried polymer film was hydrated with 2 mL of PBS at 60 °C.

The NP-[CPP]-doxo was centrifuged at 3082 × g for 10 min to remove aggregated un-encapsulated doxorubicin. To determine NP doxorubicin content, an aliquot of doxo-containing micelles was then lyophilized and dissolved in DMSO. High-performance liquid chromatography (HPLC) analysis of the diluted solution was measured and compared to standard curves for doxorubicin.

**In vitro doxorubicin release**. Doxorubicin release experiments were performed by placing 500 μL of NP-[CPP]-doxo into a Slide-A-Lyzer MINI dialysis device (Thermo Scientific) with a 10,000 molecular weight cutoff. The sample was dialyzed against 14 mL PBS and incubated at 37 °C on a platform shaker (New Brunswick Innova 40; Eppendorf) at 200 × g. At predetermined time points, the dialysis solution (release medium) was exchanged with fresh PBS. To determine doxorubicin release from irradiated NP-[CPP], nanoparticles were irradiated by 400 nm LED for 1 min at 50 mW cm$^{-2}$ at the beginning of the in vitro release study. The doxorubicin concentration in aliquots of release media was determined by HPLC ($\lambda = 233$ nm).

**Animal studies**. Healthy adult female C57BL/6 mice (6–8 weeks) weighing 19–21 g were purchased from Charles River (Wilmington, MA, USA). We complied with all relevant ethical regulations for animal testing and research. Experiments were carried out in accordance with protocols approved by Boston Children's Hospital Institutional Animal Care and Use Committee. The induction of laser-induced choroidal neovascularization was performed according to previously established protocols[39,40]. Mice were anesthetized with a mixture of ketamine (100 mg kg$^{-1}$, IP) and xylazine (6 mg kg$^{-1}$, IP). Their pupils were then dilated with a topical drop of Cyclomydril® (Alcon Laborotories, Fort Worth, TX). After pupils were dilated (3–5 min), the mouse eyes were hydrated with GenTeal® eye drops. A Micron IV imaging guided laser system (Phoenix Research Labs, Pleasanton, CA) was used to generate four laser burns (power 0.24 watts, duration 0.07 s) in each eye, in a pattern surrounding and of equal distance to the optic nerve head, while avoiding major retinal vessels. The four laser burns were imaged with the camera in Micron IV imaging system. The formation of CNV lesions was monitored by injecting anesthetized mice intraperitoneally with Fluorescein AK-FLUOR® (100 mg ml$^{-1}$, Akorn, Lake Forest, IL, USA) at 100 µg g$^{-1}$ (body weight) and taking fluorescent fundus images with a Micron IV imaging system.

**Fluorescence imaging of flat-mounted choroids**. One week after photocoagulation, mice were anesthetized with isoflurane, and their eyes were dilated with a drop of Cyclomydril®. Three minutes later, AMF-labeled nanoparticles were injected via tail vein. For mice that were treated with nanoparticles and irradiation, the mouse eyes were irradiated by 400 nm LED for 3 min at 50 mW cm$^{-2}$ immediately after (~30 s) tail vein injection. GenTeal® eye drops were then applied to hydrate the eyes. A Micron IV imaging system was used to monitor the distribution of AMF-labeled nanoparticles in the mouse fundus. Twenty-four hours later, mice were euthanized, their eyes were enucleated and cleaned in PBS. Eyes were then fixed with 4% paraformaldehyde for 1 h at room temperature, and rinsed with PBS afterwards. Under an illuminated microscope, the cornea and lens were dissected, then the entire retina was removed from the eyeball. The retina was carefully separated from the choroid, and four cuts were made to easily flatten the choroid onto a slide. After the choroids were mounted with Invitrogen antifade reagent, the slides were imaged by confocal microscopy (488 nm, Zeiss LSM 710). The image acquisition parameters were adjusted so that the fluorescence intensity from auto fluorescence is minimal. The images were digitized with a three-color camera. Volocity software (PerkinElmer, MA, USA) was used to quantify the average intensity of fluorescence in each CNV lesion.

**Biodistrubition study**. Mice were euthanized 24 h after injection of AMF-labelled nanoparticles and their organs were collected. Organs were weighed and sonicated in 500 µL 5% Triton X-100 solution (Sigma-Aldrich) on ice for 2 min, then the same volume of methanol was added to extract the AMF and another 2 min of sonication was performed. Mixtures underwent mechanical agitation for 2 min and were then centrifuged at 14,462 × g for 15 min (Microfuge 22R Centrifuge, Beckman Coulter, CA, USA). To determine the content of AMF in each tissue homogenate sample, 800 µL of the supernatant solution was transferred into a cuvette and analyzed by a fluorescence spectrometer (Agilent, CA, USA). Tissue samples from untreated mice were measured as controls for auto fluorescence, which was subtracted from the fluorescence intensity of the experimental groups. The data were divided by tissue mass (µg g$^{-1}$).

**Histology**. For ocular histology studies, enucleated eyes were embedded in OCT compound in a cryomold with its optic nerve-pupil axis oriented horizontally and frozen with liquid nitrogen vapor. Eight micrometers of cryosections of the tissue were stained with hematoxylin and eosin and assessed by light microscopy. Eyes from normal untreated mice were used as controls.

For organ histology studies, mice were euthanized after 1 week of treatment with free drug (doxorubicin), nanoparticles without drug and nanoparticles with doxorubicin. Mice injected with PBS were used as controls. Organs were collected, fixed with 10% formalin, embedded in paraffin, sectioned, and stained with hematoxylin and eosin. Histologic assessment by light microscopy was performed by a pathologist (M.M.) in a blinded fashion.

**Analysis of the CNV neovessel area**. Mouse choroids were stained with isolectin GS-IB$_4$, Alexa fluor$^{TM}$ 594 conjugate (Invitrogen), and images were obtained by a Zeiss Observer.Z1 fluorescence microscope. Images were digitized with a digital camera. ImageJ software (NIH, USA) was used to measure the total area (in µm$^2$) of CNV associated with each laser burn by an observer (Y.W.) who was blinded to the nature of the individual images. A calibration image was taken from a slide with a grating of known size. An established and constant threshold in pixels (corresponding to threshold fluorescence) was used to outline the fluorescent blood vessels and quantify the area of neovascularization. A lesion was excluded if one of the following conditions occurred, according to previously reported protocols[39,41]: (1) there was choroidal hemorrhage; (2) the lesion was linear instead of circular; (3) there was fusion of two or more lesions; (4) the lesion was the only lesion in an eye; (5) the area of the lesion was more than fivefold larger than the next biggest lesion in the same eye, or less than 1/5 the area of the next smallest lesion in the eye.

**Statistical analysis**. Statistical analysis was conducted using OriginPro software (version 8, OriginLab). All p-values were calculated by the unpaired t-test, $p < 0.05$ was considered statistically significant.

**Reporting summary**. Further information on experimental design is available in the Nature Research Reporting Summary linked to this article.

## Data availability

The data that support the findings of this study are available within the paper and its Supplementary Information. A reporting summary for this Article is available as a Supplementary Information file.

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

## Acknowledgements
This work was supported by the National Institutes of Health GM 116920 (to D.S.K.), R01EY024963 and R01EY028100 (to J.C.).

## Author contributions
Y.W., W.W. and D.S.K. conceived the study. Y.W., C.L., T.J., M.M., W.W. and E.M. performed the experiments. Y.W. and D.S.K. wrote the manuscript. All authors discussed the results, edited and approved the manuscript.

## Additional information

**Conflict of interest:** The authors declare no competing interest.

