## [Peer Review File · Nature Communications]

Reviewers' comments:

Reviewer #1 (Remarks to the Author):

The manuscript demonstrated that photo-targeted nanoparticles would reduce CNV size. However, the in vivo experiment is too simple. Figure 5 only showed data results, please add representative CNV images. In addition, evaluation CNV volume is more accurately than CNV area. If possible, please re-calculate.

For safety experiment, the authors only indicated that no tissue damage after light-triggered treatment after 24 hours.

This short-term exposure is not strong enough to show safety. Please confirm that how long the photo-targeted nanoparticles (NP-CPP) would completely washout in laser-CNV treated mice and evaluation safety time point should longer than wash out time.

The section, "therapeutic effects of phototriggered activation of nanoparticle in CNV". This part has critical problems. PLA-PEG nanoparticle is micelle with hydrophobic core. However, doxorubicin is hydrophilic and it has been used with liposomes. It is possible that PLA-PEG nanoparticle with doxorubicin is unstable. However, the author did not characterize the nanoparticle with doxorubicin well. Also, in their method, the unloaded doxorubicin could not be removed. Therefore, the authors need to show the basic characterizations of nanoparticle with doxorubicin. Especially, 1) size, 2) zeta-potential and 3) TEM. Also, if no aggregation, please say so somewhere.

1. FigureS1. This figure needs to be described more.
 - a) In the legend, "HPLC of purified [CPP]". I believe this is HPLC purification of [CPP]. If not, please describe the method to purify [CPP]. In this case, it is recommended to show HPLC data of unmodified CPP if available. If not, it is not necessary.
 - b) Figure b. Please describe the expected size of CPP and [CPP].
 - c) Figure b. To ensure no residual unmodified CPP, please extend the graph below 1700. I think unmodified CPP is around 1.7kDa and one modified CPP is around 2kDa.
 - d) Figure c. Can you put the NMR data of unmodified CPP (figure 1e)? The comparison will make the readers understand.

2. Figure 1a. In this figure, the author showed DEACM released from nanoparticle after light stimulation. However, it is possible that the large amount of DEACM be in nanoparticle. So, did you examine most of DEACM is released from nanoparticle after light stimulation?
3. What will happen in lysine after light stimulation? DEACM is conjugated with lysine. After detaching DEACM, is it still lysine? I think it will be -NH-CHO and then NH-COOH. Also, I think DEACM cleavage from lysine would require a water. However, the author said that DEACM is in the core of PLA nanoparticle, and a water is not around DEACM by fluorescence emission shift and phenyl alanine. I think this is a kind of contradiction of hypothesis. So, can you check what happen in lysine after light stimulation? If it is not lysine anymore, please describe alternative solution in the discussion as a future plan.
4. Figure 1c and Figure S2.
 - a) If available, please put TEM images of NP (PEG-PLA), NP-CPP and NP-[CPP] with irradiation. Also, please tell us those hydrodynamic diameters. Especially, I would like to know about NP-[CPP] with irradiation (whether releasing DEACM change nanoparticle properties).
 - b) Did you check zeta-potential of each nanoparticles? If you did, please put the data. One of my concern is aggregation at the physiological condition.
 - c) In the TEM image, some of nanoparticle are white, which indicates electric charge are different from others. Do you have any idea why? This is just question.
5. Figure 1f. What does "100%" mean? Is this really 100% photocleavage? I think the author assume 100% photocleavage at 120 sec since it reaches the plateau. If so, it may cause misunderstanding. Please describe this axis in the method or legend. Also, did you check it at 0 sec or assume it is 0?
6. Figure S3. Why did 40% w/w nanoparticle show the maximum uptake? As a general sense, I think the higher CPP should show higher uptake or plateau if they form nanoparticle. Did you check the nanoparticle formed at 50-60% w/w? This is just question.
7. Figure 2.
 - a) Please put non-treated HUVEC data on figure 2 a and b.
 - b) Figure 2b. Please describe "mean intensity" in the method. Is it geometric mean or arithmetic mean?
 - c) Please show FSC vs SSC of each data (maybe in the supplemental data). SSC may show the difference by nanoparticle internalization. Also, those data will tell us HUVEC not changing.
 - d) Figure 2c. Do you have the better image of NP-AMF-[CPP] + hv treated HUVEC? Since some cells showed round shape which may indicate start of detaching and the density is different from the control. If you don't have, it is not necessary to repeat.
8. Figure S4. Please put the control data (normal HUVEC) and show the statistics. From this standard deviation, irradiated HUVEC may show significant decrease of survival. Also, in this

experiment, were HUVEC incubated with Nanoparticle overnight? Did you remove the nanoparticle for MTS assay? Didn't 0.5mg/mL nanoparticle affect the absorbance?

9. Figure 3. I think 400nm 50mW/cm² LED is very strong light. Therefore, the safety will be the primary concern. Especially, the diseased eye may be susceptible to the light than the healthy eyes. Please consider more experiments in the future (not this time), such as ERG, apoptosis assay, OCT on time course (at least over 2 weeks). Also, LED laser may irradiate on the region only.

10. Line 189-198. Please describe this paragraph correctly.

a) line 189. "One work after....". I think this is "one week after".

b) In this experiment, the control (for example, non injection or PBS injection) is needed. Laser induced CNV often shows strong auto fluorescence.

c) Please reconsider the legend of a. You showed four different groups.

d) Figure 4b. CNV should be stained with other color for the normalization. Otherwise, how did you choose the lesion? By autofluorescence?

11. Do you have legends of video 1 and 2?

12. Figure S7. Please show all four groups or at least NP-AMF-[CPP] with/without irradiation. Also, please show the representative images. For example, the images at around 100 sec and 300 sec. As the author's hypothesis, irradiation may increase the fluorescence or show mild decrease than without irradiation. If so, these data make the paper strong

13. Line 196-197. Please show in vivo 8h's and 24h's images. Also, did you find any difference with/without irradiation? Because in this paragraph the author injected four groups, all four groups should be shown.

14. Line 197-198. Didn't irradiation induce the accumulation of NP-AMF-[CPP] in the retinal vasculature? If not, why?

15. Line 206-208. As mentioned above, laser induced CNV often has strong autofluorescence. In these sentences, the author assumed all of the fluorescence are due to NP. However, the majority of fluorescence may be due to autofluorescence. Therefore, please put the control data (e.g. no nanoparticle injection).

16. Line 212-214. This may indicate the weak fluorescence may be due to autofluorescence.

17. Figure S14. The error bar should be from the mean.

Reviewer #2 (Remarks to the Author):

This manuscript by Wang et al. reported a photo-targeted drug delivery strategy for choroidal neovascularization (CNV) where polymeric nanoparticles (NPs) carrying drug molecules are injected intravenously and 400 nm LED light was shone at the eyes to trigger the binding of the NPs to the CNV lesions. The authors demonstrated that, after IV injection, irradiation at the eye induced high accumulation of the NPs in the CNV lesions. They also showed that this strategy is therapeutically effective with a model drug doxorubicin encapsulated in the NPs. Photo-targeting drugs to the retina is a novel concept and the in vivo results reported here are promising. Overall, the findings are of significant importance and the experiments are well executed and support the conclusions made by the authors. The reviewer recommends the publication of this manuscript in Nature Communications after the following minor points have been addressed.

1. In the Introduction section, the authors have cited some already reported NP-delivery platforms for neovascular eye diseases, which well reflects the progress in this research area. However, the reviewer believes it will be better if the authors could further discuss the advantages of the NPs they developed in this study sepecifically compared with these previously reported NP delivery systems.
2. A brief background regarding the targeted delivery of nano-platforms (e.g., Chem Soc Rev. 2012, 41(7):2971-3010; Expert Opin Drug Deliv. 2008, 5(9):927-929; Acc Chem Res. 2011, 44(10):1123-1134; Adv Mater. 2017, 29(1): 1603276; etc.) could be helpful to improve this manuscript.
3. The authors need to add a quantitative comparison of the cell targeting abilities of 20% w/w and 40% w/w [CPP]/CPP NPs in order to better justify the use of the 20% w/w NPs.
4. The reviewer suggests the authors add the confocal images of the NP-AFM group and the NP-AMF-CPP group in Figure 2c. It is necessary to make sure these results are consistent with the flow cytometric results in Figure 2a.

Reviewer #3 (Remarks to the Author):

The manuscript describes a study in which light activatable nanoparticles were developed to enable their accumulation in choroidal neovascular lesions. To that aim, polymeric nanoparticles were functionalized with a cell penetrating TAT peptide that was molecularly 'caged'. Upon irradiation, the peptides become exposed, enhancing nanoparticle accumulation.

It seems the technology was previously reported and applied in a tumor model. Nevertheless, the technology is compelling and the current application is highly innovative.

Specific comments:

- The introduction reads as a summary. Please revise.
- The figure quality varies a lot. Please integrate Figure 1's style throughout the manuscript.
- Doxorubicin is a fluorescent molecule and can thus be quantified similar to how the fluorescent nanoparticles were quantified. It would be valuable to give this part of the study more beef and add such data.

Daniel S. Kohane M.D., Ph.D.
Professor of Anaesthesia
Director, Laboratory for Biomaterials and Drug Delivery
Senior Associate in Critical Care

Division of Critical Care Medicine
Boston Children's Hospital
300 Longwood Avenue | Bader 634
Boston, Massachusetts 02115
phone 617-355-7327 | fax 617-730-0453
daniel.kohane@childrens.harvard.edu

Laboratory for Biomaterials and Drug Delivery at Boston Children's Hospital
Enders Research Bldg | Enders 361
Boston, Massachusetts 02115
phone 617.919.2364
http://kohane.tch.harvard.edu

July 09, 2018

Reviewers' comments:

Reviewer #1 (Remarks to the Author):

The manuscript demonstrated that photo-targeted nanoparticles would reduce CNV size. However, the *in vivo* experiment is too simple.

We regret that we do not understand what the reviewer means by our *in vivo* experiment being "too simple". We followed the standard protocol for studying laser induced CNV in a mouse model (*Nature Protocols* 8, 2197, (2013)).

Figure 5 only showed data results, please add representative CNV images.

We have added representative images to Figure 5, as requested by the reviewer.

Figure 5. Treatment with NP-[CPP]-doxo in mouse CNV model. a, Cumulative doxorubicin release (as % of total amount loaded) from NP-[CPP]-doxo at 37 °C *in vitro* by dialysis, with or without irradiation (400 nm LED for 1 min at 50 mW/cm²) at t = 0. Data are means ± SD (n = 4). b, Representative isolectin GS IB₄-stained CNV images (the scale bar is 100 μm) and mean CNV lesion area from laser-induced CNV mice treated with PBS, NP-[CPP], doxo, NP-[CPP]-doxo, and NP-[CPP]-doxo with irradiation. Error bars show standard error of the mean. n = 23-28 lesions. **P* < 0.05, *P* < 0.005. (See Figure S14 for graph of individual data points and Table S1 for the standard deviations.)**

In addition, evaluation CNV volume is more accurately than CNV area. If possible, please recalculate.

Measurement of CNV area as we have done is a widely accepted metric for the evaluation of CNV lesion size. Moreover, providing volume data would involve essentially repeating the majority of the experiments since the fluorescence intensity of existing samples has faded which prevents 3D re-imaging.

For safety experiment, the authors only indicated that no tissue damage after light-triggered treatment after 24 hours.

This short-term exposure is not strong enough to show safety. Please confirm that how long the photo-targeted nanoparticles (NP-CPP) would completely washout in laser-CNV treated mice and evaluation safety time point should longer than wash out time.

We regret that we are not sure which experiment the reviewer is referring to. All our toxicity endpoints were much further out than 24 hours. We evaluated photo-toxicity to the eye by examining mouse eyes with a fundus imaging camera 48 hours after the irradiation and by histology 2 weeks after the irradiation. Organ toxicity of drug-containing nanoparticles was evaluated after 6 days of treatment. The fact that we did not observe organ toxicity after 6 days strongly suggests that toxicity is unlikely to occur, because the concentrations of the formulation and the drug are expected to decrease over time.

The section, “therapeutic effects of phototriggered activation of nanoparticle in CNV”. This part has critical problems. PLA-PEG nanoparticle is micelle with hydrophobic core. However, doxorubicin is hydrophilic and it has been used with liposomes. It is possible that PLA-PEG nanoparticle with doxorubicin is unstable. However, the author did not characterize the nanoparticle with doxorubicin well. Also, in their method, the unloaded doxorubicin could not be removed. Therefore, the authors need to show the basic characterizations of nanoparticle with doxorubicin. Especially, 1) size, 2) zeta-potential and 3) TEM. Also, if no aggregation, please say so somewhere.

We apologize that we did not highlight the fact that we used the hydrophobic free base of doxorubicin. In the original submission, we described the preparation of free-base hydrophobic doxorubicin in SI -- doxo was first converted to the hydrophobic free-base form, then encapsulated in NPs. We have now moved this paragraph to the Methods section in the

main text to make it clearer. We have also modified the Results section: “The drug doxorubicin (doxo) which inhibits CNV when injected intraocularly¹ was converted to the free-base form and then encapsulated in the nanoparticles”.

Regarding the removal of unloaded doxorubicin, as we had originally described in our Methods section, “The NP-[CPP]-doxo was centrifuged at 4000 rpm for 10 min to remove aggregated unencapsulated doxorubicin”.

As per the reviewer’s request, we have added the size, zeta potential and TEM data as Table S2 and Figure S16 in SI, and added the clarification that NPs containing doxorubicin do not aggregate.

Table S2. Hydrodynamic diameter and zeta potential of NP, NP-[CPP], NP-[CPP]+hv, NP-CPP and NP-[CPP]-doxo.

	Hydrodynamic diameter (nm)	Zeta potential (mV)
NP	18.4 ± 2.6	-6.2 ± 1.0
NP-[CPP]	19.0 ± 2.0	-4.1 ± 0.8
NP-[CPP]+ hv	20.1 ± 3.2	10.5 ± 1.4
NP-CPP	18.8 ± 2.9	9.8 ± 1.6
NP-[CPP]-doxo	29.4 ± 4.1	-1.0 ± 0.2

Figure S16. TEM images of all NPs: NP (a), NP-[CPP] (b), NP-[CPP]+hv (c), NP-CPP (d), NP-[CPP]-doxo (e). The scale bar is 50 nm.

1. FigureS1. This figure needs to be described more.

a) In the legend, “HPLC of purified [CPP]”. I believe this is HPLC purification of [CPP]. If not, please describe the method to purify [CPP]. In this case, it is recommended to show HPLC data of unmodified CPP if available. If not, it is not necessary.

The HPLC of purified [CPP] was to demonstrate the presence of a single peak in the chromatogram. The purification method was described in SI in the original submission; we have now moved it to the main text:

“The mixture underwent mechanical agitation for 24 hours, then was separated by HPLC (C4 column, “H8” 214TP52, 2.1X150mm, from Vydac, Hesperia, CA). The mobile phase consisted of solvent A (0.05% TFA in H₂O) and B (0.043 % TFA, 80 % ACN in H₂O). 10% solvent B was used as the mobile phase from 0 to 10min, then the percentage of B in the mobile phase was linearly increased to 100% from 10 to 55min. The flow rate was 0.3 mL/min.”

b) Figure b. Please describe the expected size of CPP and [CPP].

We have added the molecular weights of CPP and [CPP] in the Figure S1 caption (see below). The MW of CPP is 1746.1 Da and the MW of [CPP] is 2292.6 Da.

c) Figure b. To ensure no residual unmodified CPP, please extend the graph below 1700. I think unmodified CPP is around 1.7kDa and one modified CPP is around 2kDa.

We have replaced the MALDI spectrum with the electrospray ionization mass spectroscopy (ESI-MS) spectrum, because in MALDI MS, the laser source could lead to uncaging of a small fraction of [CPP]. We have updated the Figure S1 caption to include the method (see below). All the peaks in the ESI-MS spectrum are m/z peaks from [CPP] (m/z:2292.2, the 2M masses are due to dimerization of SH in the ESI chamber, which was confirmed by chromatogram of [CPP] and dimerized [CPP]). We have also included the following clarification in the Methods section: "The mass spectrum of [CPP] shows only peaks corresponding to CPP modified with two photo-caging groups (Figure S1)."

Figure S1. Characterization of [CPP]. a, HPLC of purified [CPP], monitored at 210nm (blue line), 280nm (red line) and 385nm (green line). b, Electrospray ionization mass spectroscopy (ESI-MS) spectrum of [CPP](m/z = 2292.2), using positive scan mode on an Agilent 6130 Single Quad LCMS instrument. MW of unmodified CPP is 1746.1. c, NMR spectrum of [CPP], in DMSO-d₆.

d) Figure c. Can you put the NMR data of unmodified CPP (figure 1e)? The comparison will make the readers understand.

The top spectrum of Figure 1e is the NMR of unmodified CPP. We have also added the full NMR spectrum of unmodified CPP in the SI.

Figure S15. Full NMR spectrum of unmodified CPP.

2. Figure1a. In this figure, the author showed DEACM released from nanoparticle after light stimulation. However, it is possible that the large amount of DEACM be in nanoparticle. So, did you examine most of DEACM is released from nanoparticle after light stimulation?

In response to the reviewer’s comment, we have added the following clarification in the Methods section. “After irradiation, NPs were separated from the solution by centrifugal filtration and then dissolved in acetonitrile and analyzed by HPLC. There was no detectable DEACM in the NPs by HPLC analysis.”

3. What will happen in lysine after light stimulation? DEACM is conjugated with lysine. After detaching DEACM, is it still lysine? I think it will be -NH-CHO and then NH-COOH. Also, I think DEACM cleavage from lysine would require a water. However, the author said that DEACM is in the core of PLA nanoparticle, and a water is not around DEACM by fluorescence emission shift and phenyl alanine. I think this is a kind of contradiction of hypothesis. So, can you check what happen in lysine after light stimulation? If it is not lysine anymore, please describe alternative solution in the discussion as a future plan.

After the photocleavage reaction, it is still lysine. The mechanism of the photo-cleavage reaction has been described, for example in *J. Org. Chem.* 2002, 67, 703-710 and *Chem. Rev.* 2013, 113, 119–191. The reaction mechanism is shown below. We have added this as Figure S20 in the SI.

Figure S20. The photocleavage reaction ([CPP] to CPP) mechanism.¹⁻²

The photocleavage does not require a water molecule, because it goes through the radical pathway (as drawn above and shown in the references). However, the formation of DEACM-OH does require a hydroxyl group from the solvent (H_2O). The fluorescence emission shift experiment referred to by the reviewer indeed shows a more hydrophobic environment. However that does not mean complete absence of water molecules in the PLA core of our micelles (vacuum between PLA polymer chains would be highly thermodynamically unfavorable). We have added a sentence in the Methods section: “The mechanism of the photocleavage reaction is shown in Figure S20 (Supporting Information).”

4. Figure 1c and Figure S2.

a) If available, please put TEM images of NP (PEG-PLA), NP-CPP and NP-[CPP] with irradiation. Also, please tell us those hydrodynamic diameters. Especially, I would like to know about NP-[CPP] with irradiation (whether releasing DEACM change nanoparticle properties).

We have added the TEM images of all NPs (Figure S16) to the SI, and their hydrodynamic diameters (Table S2). Irradiation did not change the diameter or the morphology of NPs.

Figure S16. TEM images of all NPs: NP (a), NP-[CPP] (b), NP-[CPP]+hv (c), NP-CPP (d), NP-[CPP]-doxo (e). The scale bar is 50 nm.

Table S2. Hydrodynamic diameter and zeta potential of NP, NP-[CPP], NP-[CPP]+hv, NP-CPP and NP-[CPP]-doxo.

	Hydrodynamic diameter (nm)	Zeta potential (mV)
--	----------------------------	---------------------

NP	18.4 ± 2.6	-6.2 ± 1.0
NP-[CPP]	19.0 ± 2.0	-4.1 ± 0.8
NP-[CPP]+ hv	20.1 ± 3.2	10.5 ± 1.4
NP-CPP	18.8 ± 2.9	9.8 ± 1.6
NP-[CPP]-doxo	29.4 ± 4.1	-1.0 ± 0.2

b) Did you check zeta-potential of each nanoparticles? If you did, please put the data. One of my concern is aggregation at the physiological condition.

We have added the zeta-potential of all nanoparticles (Table S2), and added in the Methods section:

“The zeta potential of nanoparticles was measured by a PALS Zeta Potential Analyzer (Brookhaven Instruments Corp.). 0.1 mg/mL NPs (800 μL) in 5 mM phosphate buffer was tested at 25°C for at least 4 times. When incubated with 10% FBS solution, the NPs did not form aggregates over 48 hrs.”

Table S2. Hydrodynamic diameter and zeta potential of NP, NP-[CPP], NP-[CPP]+hv, NP-CPP and NP-[CPP]-doxo.

	Hydrodynamic diameter (nm)	Zeta potential (mV)
NP	18.4 ± 2.6	-6.2 ± 1.0
NP-[CPP]	19.0 ± 2.0	-4.1 ± 0.8
NP-[CPP]+ hv	20.1 ± 3.2	10.5 ± 1.4
NP-CPP	18.8 ± 2.9	9.8 ± 1.6
NP-[CPP]-doxo	29.4 ± 4.1	-1.0 ± 0.2

c) In the TEM image, some of nanoparticle are white, which indicates electric charge are different from others. Do you have any idea why? This is just question.

We think the bright white spots are non-covered grid or thinly deposited staining agent (uranyl acetate).

5. Figure 1f. What does “100%” mean? Is this really 100% photocleavage? I think the author assume 100% photocleavage at 120 sec since it reaches the plateau. If so, it may cause misunderstanding. Please describe this axis in the method or legend. Also, did you check it at 0 sec or assume it is 0?

Yes, 100% means full photocleavage. After 120 seconds of irradiation, NPs were separated from the solution by centrifugal filtration and then dissolved in acetonitrile and analyzed by HPLC. There was no detectable DEACM-OH in the NPs by HPLC analysis. Similarly, at time 0, there was no detectable DEACM-OH in the NPs by HPLC.

We have added this information in the Methods section:

“At time 0, there was no detectable DEACM-OH in the NPs by HPLC. After irradiation, NPs were separated from the solution by centrifugal filtration and then dissolved in acetonitrile and analyzed by HPLC. There was no detectable DEACM-OH in the NPs HPLC analysis.”

6. Figure S3. Why did 40% w/w nanoparticle show the maximum uptake? As a general sense, I think the higher CPP should show higher uptake or plateau if they form nanoparticle. Did you check the nanoparticle formed at 50-60% w/w? This is just question.

We did study nanoparticles formed with 50% and 60% CPP-PEG-PLA. We incubated these two groups of NPs with 10% FBS solution, and saw much more obvious precipitation after 48 hours, compared to NPs formed from 20% or 40% CPP-PEG-PLA. (Our hypothesis is that such NPs would absorb more protein on their surface in cell culture medium, which could impede their cellular uptake.) Consequently, they were not used in subsequent experiments.

7. Figure 2.

a) Please put non-treated HUVEC data on figure 2 a and b.

We have added the non-treated HUVEC group in Figure 2 a and b. Thank you for the suggestion.

Figure 2. Light-triggered cell uptake of nanoparticles a, Representative flow cytometry of FITC fluorescence within HUVEC cells treated with different or without nanoparticle. b, Quantitation (mean of four median values of fluorescence intensity) of flow cytometric analyses (such as the one in panel A) of HUVEC uptake of nanoparticles. Data are means \pm SD ($n = 4$). * $P < 0.001$. c, Representative confocal microscopic images of HUVEC uptake of nanoparticles. The scale bar is 20 μ m.**

b) Figure 2b. Please describe “mean intensity” in the method. Is it geometric mean or arithmetic mean?

The data in Figure 2b are arithmetic means of flow cytometric data. The latter data are medians, as provided by the flow cytometry software. (Presumably medians are used because flow cytometry data are often not normally distributed.)

“Quantitation (mean of four median values of fluorescence intensity) of flow cytometric analyses (such as the one in panel A) of HUVEC uptake of nanoparticles. Data are means \pm SD ($n = 4$). *** $P < 0.001$.”

c) Please show FSC vs SSC of each data (maybe in the supplemental data). SSC may show the difference by nanoparticle internalization. Also, those data will tell us HUVEC not changing.

We have added all the FSC vs SSC data in SI. As shown in these figures, the size and granularity of HUVECs remain the same for all groups.

Figure S17 is called out in the text as follows: “The size and granularity of cells remained the same for all groups. (Figure S17)”

Figure S17. Representative FSC (Forward SCatter)-SSC (Side SCatter) (from flow cytometry) data. The purple color labels cells that are not FITC positive, while the green color labels FITC-positive cells. The size (from FSC) and granularity (from SSC; this metric would detect breakdown and/or aggregation of cells) of HUVECs remained the same for all groups.

d) Figure 2c. Do you have the better image of NP-AMF-[CPP] + hv treated HUVEC? Since some cells showed round shape which may indicate start of detaching and the density is different from the control. If you don't have, it is not necessary to repeat.

Cells were exposed to PBS in order to do the imaging, which we think is the reason that they might start to detach. Consequently these were the best images we have. However, both

groups of cells were able to recover to a healthy and adherent state after they were put back in the incubator and supplemented with fresh medium.

8. Figure S4. Please put the control data (normal HUVEC) and show the statistics. From this standard deviation, irradiated HUVEC may show significant decrease of survival. Also, in this experiment, were HUVEC incubated with Nanoparticle overnight? Did you remove the nanoparticle for MTS assay? Didn't 0.5mg/mL nanoparticle affect the absorbance?

NPs were incubated overnight, and were removed before the MTS assay. We have added this clarification in the main text in the Results section:

“HUVECs were exposed to irradiation (400 nm LED for 1 min at 50 mW/cm²), or incubated with 0.5 mg/mL NP-[CPP] overnight with or without irradiation (1 min, at the beginning of incubation). NPs were removed before the MTS assay. Non-treated HUVECs were used as controls. All three groups showed high cell viability (Fig. S4).”

We have added the control data and statistics to Figure S4.

Figure S4. Survival rates determined by MTS assay of HUVECs after irradiation (400 nm at 50 mW/cm² for 1 min at the beginning of incubation) and/or overnight incubation with 0.5 mg/mL NP-[CPP], with non-treated HUVECs as controls. Data are means ± SD (n = 4).

9. Figure 3. I think 400nm 50mW/cm² LED is very strong light. Therefore, the safety will be the primary concern. Especially, the diseased eye may be susceptible to the light than the healthy eyes. Please consider more experiments in the future (not this time), such as ERG, apoptosis assay, OCT on time course (at least over 2 weeks). Also, LED laser may irradiate on the region only.

We appreciate the advice and will certainly consider the suggested safety evaluation experiments in our future study.

10. Line 189-198. Please describe this paragraph correctly.

a) line 189. “One work after....”. I think this is “one week after”.

Thank you. We have corrected the error.

b) In this experiment, the control (for example, non injection or PBS injection) is needed. Laser induced CNV often shows strong auto fluorescence.

We have added the control eye to the figure. It is true that laser-induced CNV has auto fluorescence at the wavelength we use for AMF measurement. However, we adjusted the image acquisition parameters so that the fluorescence intensity from auto fluorescence is minimal. We have added this clarification in the Methods section: “After the choroids were mounted with Invitrogen antifade reagent, the slides were imaged by confocal microscopy (488nm, Zeiss LSM 710). The image acquisition parameters were adjusted so that the fluorescence intensity from auto fluorescence is minimal.”

Figure 4. Light-triggered targeting of CNV *in vivo* a, Representative (of 8) fluorescent images of flat-mounted choroid 24 hours after injection with NPs. The scale bar is 100 μm . b, Quantification of the intensity of fluorescent neovessels from images in panel (a), normalized by the lesion size. Data are means \pm SD (n = 8) *** $P < 0.001$.

c) Please reconsider the legend of a. You showed four different groups.

Thank you for the suggestion, we have changed the figure Caption to “Representative (of 8) fluorescent images of flat-mounted choroid 24 hours after injection with NPs. The scale bar is 100 μm .”

d) Figure 4b. CNV should be stained with other color for the normalization. Otherwise, how did you choose the lesion? By autofluorescence?

We used fluorescence (from NPs and auto-fluorescence) to determine the area (after adjusting contrast and curves of the image). We have confirmed that the area determined by this approach is the same as the area determined by isolectin staining. We did not use isolectin staining itself to define lesions because, as shown in the figure below, isolectin

staining greatly increases the background signal in the AMF channel. Since this background signal varies between different flat-mount slides, it would be unreliable to use isolectin-stained slides for fluorescence quantification. We therefore used unstained slides for fluorescence quantification.

Isolectin channel

AMF channel

Overlay

11. Do you have legends of video 1 and 2?

We have added legends of video 1 and 2 in SI.

Video 1: Fluorescence of mouse fundus 1.5-3 min after IV injection of NP-AMF.

Video 2: Fluorescence of mouse fundus 4-5 min after IV injection of NP-AMF.

12. Figure S7. Please show all four groups or at least NP-AMF-[CPP] with/without irradiation. Also, please show the representative images. For example, the images at around 100 sec and 300 sec. As the author's hypothesis, irradiation may increase the fluorescence or show mild decrease than without irradiation. If so, these data make the paper strong

We have added representative images of fluorescence angiograms to Figure S7. We did not take pictures at 100 s in the irradiated groups because the animals were being irradiated at the time. Therefore, there would be no data to compare a 300s timepoint to, in those groups.

Figure S7. Fluorescence intensity in the retina. Quantification in a representative single animal of the average fluorescent intensity in (a) retinal blood vessels and (b) laser-induced lesions, after IV injection of fluorescently labeled nanoparticles. (c, d) Representative images at (c) 100s and (d) 300s after NP-AMF injection.

13. Line 196-197. Please show in vivo 8h's and 24h's images. Also, did you find any difference with/without irradiation? Because in this paragraph the author injected four groups, all four groups should be shown.

We have added the images. We did not see significant differences between groups due to the low resolution and sensitivity of the in vivo imaging method.

Figure S18. In vivo fluorescence images of mouse fundus 8hrs (a-d) and 24 hrs (e-h) after IV injection of nanoparticles. Faint fluorescence in blood vessels and lesions can be seen at 8 h.

14. Line 197-198. Didn't irradiation induce the accumulation of NP-AMF-[CPP] in the retinal vasculature? If not, why?

As evidenced by Figure S8, NPs did not accumulate in the retinal vasculature. Our hypothesis is that the vascular leakiness due to CNV is necessary for enhancing NP accumulation.

15. Line 206-208. As mentioned above, laser induced CNV often has strong autofluorescence. In these sentences, the author assumed all of the fluorescence are due to NP. However, the majority of fluorescence may be due to autofluorescence. Therefore, please put the control data (e.g. no nanoparticle injection).

We have added the control data, which show that autofluorescence is a relatively small contributor to total fluorescence.

Figure 4. Light-triggered targeting of CNV *in vivo* a, Representative (of 8) fluorescent images of flat-mounted choroid 24 hours after injection with NPs. The scale bar is 100 μm . b, Quantification of the intensity of fluorescent neovessels from images in panel (a), normalized by the lesion size. Data are means \pm SD ($n = 8$) *** $P < 0.001$.

16. Line 212-214. This may indicate the weak fluorescence may be due to autofluorescence.

As stated above, the autofluorescence is minimized with the imaging parameters we used.

17. Figure S14. The error bar should be from the mean.

Since we have the error bar in Figure 5, we removed the error bar from this figure (S14) to make it clearer.

Reviewer #2 (Remarks to the Author):

This manuscript by Wang et al. reported a photo-targeted drug delivery strategy for choroidal neovascularization (CNV) where polymeric nanoparticles (NPs) carrying drug molecules are injected intravenously and 400 nm LED light was shone at the eyes to trigger the binding of the NPs to the CNV lesions. The authors demonstrated that, after IV injection, irradiation at the eye induced high accumulation of the NPs in the CNV lesions. They also showed that this strategy is therapeutically effective with a model drug doxorubicin encapsulated in the NPs. Photo-targeting drugs to the retina is a novel concept and the in vivo results reported here are promising. Overall, the findings are of significant importance and the experiments are well executed and support the conclusions made by the authors. The reviewer recommends the publication of this manuscript in Nature Communications after the following minor points have been addressed.

We thank the reviewer for these positive comments.

1. In the Introduction section, the authors have cited some already reported NP-delivery platforms for neovascular eye diseases, which well reflects the progress in this research area. However, the reviewer believes it will be better if the authors could further discuss the advantages of the NPs they developed in this study sepecifically compared with these previously reported NP delivery systems.

We have already addressed the difference between our nanoparticles and those previously used for ophthalmic drug delivery. The key point, we believe, is not the nanoparticle per se, but the photo-triggering.

In our original submission, we wrote that “DDSs that enable drug delivery to the back of the eye³ are administered locally by intravitreal injection, or systemically. Systemic DDS can reach diseased sites due to the leaky vasculature in neovascular eye diseases⁴⁻⁵ or by targeting the ligand-modified DDS to specific antigens.⁶⁻⁹ Such targeting is impeded by variability in the expression of ligand receptor at the diseased site and, and by the basal expression of certain target antigens (e.g., endoglin, integrin) in normal tissue.¹⁰”

We have stated the advantages of our strategy are the administration is less invasive (compared with intravitreal injection) and it would enable targeted drug delivery with higher resolution. “Externally triggered targeting can enable drug delivery with high spatial and temporal resolution.¹¹⁻¹² Light is especially attractive as the energy source for targeting the retina, since the eye is designed to admit light. We and others have demonstrated the possibility of using light to control targeting of nanoparticles to cells and tumors.¹³⁻¹⁶ Here we designed a system whereby nanoparticles (NPs) are administered intravenously, and are converted to a tissue-targeting state only upon irradiation in the eye (Scheme 1). Our strategy would allow the targeted accumulation of drug to be triggered locally at the back of the eye, while minimizing drug deposition at off-target sites in healthy parts of the eye and in the rest of the body.”

2. A brief background regarding the targeted delivery of nano-platforms (e.g., Chem Soc Rev. 2012, 41(7):2971-3010; Expert Opin Drug Deliv. 2008, 5(9):927-929; Acc Chem Res. 2011, 44(10):1123-1134; Adv Mater. 2017, 29(1): 1603276; etc.) could be helpful to improve this manuscript.

We have added these references.

3. The authors need to add a quantitative comparison of the cell targeting abilities of 20% w/w and 40% w/w [CPP]/CPP NPs in order to better justify the use of the 20% w/w NPs.

We have added this information to the text:

“From the flow cytometric analyses (Figure S3), the ratio of cell-associated NP fluorescence in cells exposed to NPs with 20% CPP to NPs with 20% [CPP] was 4.7; that of NPs with 40% CPP to those with 40% [CPP] was 3.3.”

4. The reviewer suggests the authors add the confocal images of the NP-AMF group and the NP-AMF-CPP group in Figure 2c. It is necessary to make sure these results are consistent with the flow cytometric results in Figure 2a.

We have added the confocal images of cells incubated with NP-AMF group and the NP-AMF-CPP group in SI. We also added a sentence in the Results section of the paper: “Cells incubated with NP-AMF-CPP showed strong fluorescence, and those incubated with NP-AMF showed negligible fluorescence. (Figure S19)”

Figure S19. Representative confocal microscopic images of HUVEC uptake of nanoparticles. The scale bar is 100 μ m.

Reviewer #3 (Remarks to the Author):

The manuscript describes a study in which light activatable nanoparticles were developed to enable their accumulation in choroidal neovascular lesions. To that aim, polymeric nanoparticles were functionalized with a cell penetrating TAT peptide that was molecularly 'caged'. Upon irradiation, the peptides become exposed, enhancing nanoparticle accumulation.

It seems the technology was previously reported and applied in a tumor model. Nevertheless, the technology is compelling and the current application is highly innovative.

Specific comments:

- The introduction reads as a summary. Please revise.

We share the reviewer's distaste for Introduction segments that read as summary. In this case, the text actually is not a summary of results, it is an introduction to the logic of the experiments to follow and makes the Results section easier to read. There was one sentence, though, which was perhaps too summary-like, (in bold below), that we have changed.

Externally triggered targeting can enable drug delivery with high spatial and temporal resolution.¹¹⁻¹² Light is especially attractive as the energy source for targeting the retina, since the eye is designed to admit light. We and others have demonstrated the possibility of using light to control targeting of nanoparticles to cells and tumors.¹³⁻¹⁶ Here we designed a system whereby nanoparticles (NPs) are administered intravenously, and are converted to a tissue-targeting state only upon irradiation in the eye (Scheme 1). Our strategy would allow the targeted accumulation of drug to be triggered locally at the back of the eye, while minimizing drug deposition at off-target sites in healthy parts of the eye and in the rest of the body.

We designed photo-targeted nanoparticles formed by self-assembly of a chemically modified poly(ethylene oxide)-poly(D,L-lactic acid) (PEG-PLA) block copolymer (Figure 1a). The nanoparticles' surfaces were modified with Tat-C (48-57) cell penetrating peptide (CPP) as the targeting moiety due to its high cellular uptake¹⁷. The biological activity of the peptide was reversibly inactivated by covalent binding to a photocleavable caging group, 7-(diethylamino) coumarin-4-yl]methyl carboxyl (DEACM), which was selected for its high photocleavage efficiency and relatively long (400 nm) absorption wavelength (low phototoxicity).¹⁸ Upon irradiation, the caging group would be removed by bond cleavage so that the peptide could readily bind to nearby cells. The DEACM-CPP functionalized nanoparticles could then enhance the accumulation of drugs at the diseased site and minimize off-target drug delivery. Importantly, this approach would obviate the need for intraocular injections with their attendant risks, and improve patient compliance.

- The figure quality varies a lot. Please integrate Figure 1's style throughout the manuscript.

We are not sure what the difference is that the reviewer is alluding to. We have made some modifications to the figures to provide uniformity. However, please note that Figure 1 is different from the other figures. It shows a schematic, chemical structures, and data. The other figures generally show data on a fairly specific topic.

- Doxorubicin is a fluorescent molecule and can thus be quantified similar to how the fluorescent nanoparticles were quantified. It would be valuable to give this part of the study more beef and add such data.

We appreciate this suggestion, and have added a discussion of the matter in the manuscript: “The retention and clearance of doxorubicin in the CNV lesions and in mouse organs could be monitored by fluorimetry (of doxorubicin) of organ extracts at different time points following intravenous injection of doxorubicin-containing NPs. Such information may be helpful in optimizing the dosing regimen for treating CNV by the intravenous route, especially if further modification of the phototargeted NPs provided more extended release of the encapsulated therapeutics which would allow for less frequent dosing.”

We agree that this would be valuable in our future work where we would optimize the NP formulation to provide more extended release of the encapsulated therapeutics and the dosing regimen accordingly.

With regards,

Daniel S. Kohane

Reviewers' comments:

Reviewer #1 (Remarks to the Author):

The authors have made significant revisions and the paper is greatly strengthened. I recommend acceptance.

Reviewer #2 (Remarks to the Author):

The authors have performed additional experiments to support their conclusion and all the concerns have been addressed. The reviewer supports the acceptance of this revised manuscript.

Reviewer #3 (Remarks to the Author):

Unfortunately, the authors did marginally improve the quality of the figures

- The figure quality varies a lot. Please integrate Figure 1's style throughout the manuscript.

We are not sure what the difference is that the reviewer is alluding to. We have made some modifications to the figures to provide uniformity. However, please note that Figure 1 is different from the other figures. It shows a schematic, chemical structures, and data. The other figures generally show data on a fairly specific topic.

Reviewer: Figure 1 is a compelling multipanel figure, while the other figures look like panels that are individually presented. Moreover, overall the quality of the figures is poor. Figure 5A looks like it was generated on Windows 1.01 computer.

Boston Children's Hospital
Department of Anesthesiology, Perioperative and Pain Medicine

HARVARD MEDICAL SCHOOL
Department of Anesthesia

Daniel S. Kohane M.D., Ph.D.

Professor of Anaesthesia

Director, Laboratory for Biomaterials and Drug Delivery

Senior Associate in Critical Care

Division of Critical Care Medicine

Boston Children's Hospital

300 Longwood Avenue | Bader 634

Boston, Massachusetts 02115

phone 617-355-7327 | fax 617-730-0453

daniel.kohane@childrens.harvard.edu

Laboratory for Biomaterials and Drug Delivery

at Boston Children's Hospital

Enders Research Bldg | Enders 361

Boston, Massachusetts 02115

phone 617.919.2364

<http://kohane.tch.harvard.edu>

Jan 18, 2019

Reviewers' comments:

Reviewer #1 (Remarks to the Author):

The authors have made significant revisions and the paper is greatly strengthened. I recommend acceptance.

Thank you.

Reviewer #2 (Remarks to the Author):

The authors have performed additional experiments to support their conclusion and all the concerns have been addressed. The reviewer supports the acceptance of this revised manuscript.

Thank you.

Reviewer #3 (Remarks to the Author):

Unfortunately, the authors did marginally improve the quality of the figures

- The figure quality varies a lot. Please integrate Figure 1's style throughout the manuscript. We are not sure what the difference is that the reviewer is alluding to. We have made some modifications to the figures to provide uniformity. However, please note that Figure 1 is different from the other figures. It shows a schematic, chemical structures, and data. The other figures generally show data on a fairly specific topic.

Reviewer: Figure 1 is a compelling multipanel figure, while the other figures look like panels that are individually presented. Moreover, overall the quality of the figures is poor. Figure 5A looks like it was generated on Windows 1.01 computer.

We have consulted with the editorial office, who asked us to incorporate Scheme 1 into Figure 1 but leave the rest unchanged.

Figure 1. Preparation and characterization of phototargeted nanoparticles. **a**, Phototargeting intravenously-administered nanoparticles to choroidal neovascularization. **b**, Schematic of light-triggered activation of the nanoparticle. **c**, Synthesis of the polymer chain functionalized with caged CPP ([CPP]). **d**, Transmission electron microscopy (TEM) image of NP-[CPP]. **e**, Fluorescence emission spectra of NP-[CPP] and NP-[CPP] irradiated for 1 min (50 mW cm^{-2} , 400 nm) in PBS. **f**, ¹H NMR spectra of free CPP and different nanoparticles in D₂O, with the signature phenylalanine proton peaks highlighted in the blue rectangle. NP-CPP is the nanoparticle formed from CPP-PEG-PLA and mPEG-PLA (1:4 weight ratio). Irradiation was with a 400 nm LED for 1 min at 50 mW cm^{-2} . **g**, Photocleavage of NP-[CPP] in PBS (0.5 mg mL^{-1}), as determined by HPLC (detected at 390 nm absorbance), after continuous irradiation (50 mW cm^{-2} , 400 nm) (data are means \pm SD; n = 4).

With regards,

D. Kohane

Daniel S. Kohane